# Nutritional quality of selected commercially available seed oils and effect of storage conditions on their oxidative stability

Iqbal Ahmed[1], Shahzad Ali Shahid Chatha[1], Neelam Iftikhar[1], Muhammad Furqan Farooq[1], Hira Zulfiqar[1,2], Shafaqat Ali[3,4]*, Syed Makhdoom Hussain[5], Muhammed Ali Alshehri[6], Khalid A. Al-Ghanim[7], Abdullah Ijaz Hussain[1]*

1 Department of Chemistry, Government College University Faisalabad, Faisalabad, Pakistan,
2 Department of Chemistry "Giacomo Ciamician" University of Bologna, Bologna, Italy, 3 Department of Environmental Science, Government College University Faisalabad, Faisalabad, Pakistan, 4 Department of Biological Sciences and Technology, China Medical University, Taichung, Taiwan, 5 Department of Zoology, Government College University Faisalabad, Faisalabad, Pakistan, 6 Faculty of Sciences, Department of Biology, University of Tabuk, Tabuk, Saudi Arabia, 7 Department of Zoology, College of Science, King Saud University, Riyadh, Saudi Arabia

* abdullahijaz@gcuf.edu.pk (AIH); shafaqataligill@yahoo.com (SA)

**Data Availability Statement:** All relevant data are within the manuscript and its Supporting Information files.

## Abstract

The primary objective of this research was to investigate nutritional composition of soybean, canola, cottonseed, palm and rapeseed oils under and the effect of storage conditions on their oxidative stability. Nutritional quality of selected seed oils was determined in term of fatty acids, tocopherols and tocotrienols compositions, total phenolic, total flavonoids and mineral contents. High resolution gas chromatography (HR-GC) analysis showed the presence of saturated, monounsaturated and polyunsaturated fatty acids having range from 9.21–43.25, 27.01–58.87 and 29.23–57.75 g/100g, respectively in all the oils. High performance liquid chromatography (HPLC) analysis revealed that γ–tocopherol was the major tocopherol followed by α-tocopherol in most of the oils. Spectrophotometric analysis showed that the total phenolic contents were 2.84–14.44 mg/g of oil, measured as gallic acid equivalent and total flavonoid contents were 0.44–1.56 mg/g of oil, measure as quercetin equivalent. Inductively coupled plasma-optical emission spectrophotometer analysis revealed that Mg, Fe and Mn were present in higher concentration ranging from 57.14–114.85, 126.87–460.06 and 106.85–538.39 μg/ml respectively. For study the effect of various storage conditions on the oxidation parameters, free fatty acid, peroxide value, para-anisidine value, conjugated dienes and trienes values were determined and ranging from 0.48–1.65, 10.65–40.15 meq/kg, 9.98–33.30, 8.74–28.41 and 3.86–15.02, respectively after 90 days storage. Statistical analysis revealed that various storage conditions exerted significant ($p \leq 0.05$) effect on the oxidative stability of selected oils to different extent.

**Funding:** The authors express their sincere appreciation to the Researchers Supporting Project Number (RSP2024R48), King Saud University, Riyadh, Saudi Arabia.

**Competing interests:** The authors declare no conflict of interest.

## Introduction

Seed crops are fundamental to global agriculture and food security, serving as the basis for many of the world's staple foods and edible oils. Edible oil is an important component of most of the Asian food recipes and the quality of edible oil is mainly dependent upon its resistance towards oxidation reactions [1]. The process of oxidation of edible oil is a major problem in food industries due to lack of technology and storage plan which cause unpleasant smell as well as taste and could decline the quality of food product [2]. The oxidation of oil not only decreases the shelf life of lipid containing food products but also their further use creates health issues due to the formation of oxides of cholesterols that can cause the formation of plaque in blood vessels, consequently risk of cardiovascular diseases increases [3].

During the whole process from the manufacturing to storage it is quite important to monitor the lipid oxidation rate and extent to which it may occur. Edible oils mostly consisted of triacylglycerol mainly polyunsaturated fatty acids (PUFAs) which are susceptible towards any type of oxidation like auto-oxidation, thermal oxidation and photosensitized oxidation [4]. The degree or extent of oxidation can be determined by using different methods which determined the quantity of intermediates or products formed during specific stage of reaction [5]. In addition to triacylglycerols they also entail other minor constituents which can influence oxidation as well as quality of these oils. Some of these are helpful in preserving quality of these oils as they behave an antioxidant (tocopherol, phenolics carotenoids etc.) and retard oxidation [6]. On the other hand, some other minor components in edible oil are prone to oxidative deterioration and result in quality loss such as metal ions as they behave as catalyst and cause further oxidation [7].

In addition to nutritional composition of edible oils, storage conditions are important factors that affect the rate of oxidation which directly influence their quality [8]. Storage of edible oils under harsh or improper environmental conditions may increase the rate of deterioration of lipid containing foods. Various factors including temperature, presence of light, contact with oxygen and types of packing materials are responsible for the rate of oxidation. Primary products formed during oxidation of oils are hydroperoxides which on further oxidation may result in the formation of secondary products [9]. These include aldehydes, ketones and other small molecules which may cause food poisoning [9]. The process of oxidation also determines storage time during which lipids containing foods could remain safe and after this period could be harmful [10].

Various studies were conducted during the past which elaborate the impacts of different factors on oxidative stability of vegetable oils [11,12]. Elevation of temperature at storage places can considerably reduce oxidative stability of oils but, the magnitude of temperature influencing this could be dissimilar for different types of oils [11]. The increase in the storage temperature, increase the rate of oxidation in seed oils and every 10 ˚C escalation of temperature can roughly increase oxidative deterioration up to two folds because rate of reaction of oxygen with fatty acids become almost double [12]. Furthermore, rate of secondary oxidation products formation also increases as the temperature rises because increment in temperature not only results in faster autoxidation but also faster degradation of primary products [13]. In addition to temperature, presence of light may cause negative impact on oxidative stability of oil because presence of light results in photo-oxidation of oil [14]. But the influence of light on oxidative stability may be less when there is elevated temperature at storage places and may result in fewer variations in nutritional profiling of oil [15].

The major aim to conduct this research work was to study the effect of different storage conditions i.e. room temperature and accelerated temperature and in the presence and absence of light on the oxidative parameters of soybean, canola, cottonseed, palm and rapeseed oils.

Furthermore, the variation in the fatty acid and tocols composition, phenolic contents and mineral contents of selected oils were studied to correlate with the oxidative stability.

## Materials and methods

### Collection of oils

Samples of refined bleached and deodorized (RBD) soybean, canola, cottonseed, palm and rapeseed oils were collected from Kashmir United Oil Industry Pvt. Ltd., Faisalabad, Pakistan. These oil samples were collected in 1000 ml transparent polyethylene terephthalate (PET) bottles and were transferred to the lab and kept in refrigerator before storage. All the chemicals and reagent used for this research work were of analytical grade and purchased from Merck (Darmstadt, Germany).

### Analysis of collected samples

The collected RBD oil samples were analyzed to determine fatty acid compositions, tocopherols and tocotrienols contents, total phenolic and flavonoid contents and metal ions.

### Fatty acid composition analysis

Composition of the selected oils was analysed by gas chromatography coupled with flame ionization detector (GC-FID, Perkin Elmer). IUPAC standard protocol was applied which based upon derivatization of fatty acids into fatty acid methyl esters (FAMEs) as reported by Anwar et al. [16,17]. To prepare FAMEs a weighed amount of 0.2 g oil sample, 0.1 g KOH and 30 ml methanol were added to a round bottom flask. Reflux it for 30 minutes. After this solution was kept cooling down and was transferred to separating funnel and 10 ml of pure hexane was also added. After this whole solution mixture was gently shaken and allowed to separate into two layers. Hexane layer was collected quite carefully and further washed with distilled water two to three times. The hexane solution was dried and filtered over anhydrous sodium sulphate. For analysis, dry solvent-free methyl esters were preserved in GC vials in the freezer.

### Tocopherol and tocotrienol contents analysis

Tocol contents were determined by using HPLC (Perkin Elmer) in accordance with Current Protocols in Food Analytical Chemistry (CPFA) method, as reported [16,17]. Briefly, weighed 0.1 g of oil, 0.05 g of ascorbic acid, 5 ml of 90.2% ethyl alcohol and 0.5 ml of 80% potassium hydroxide was added in test tube and was vortexed for about 30 to 40s. After this these tubes were placed into ice bath for 4 to 5 minutes. After this 3 ml of deionized water and 5 ml pure hexane was added into the test tube, this obtained solution mixture was vortexed for half minute followed by centrifugation at 4500 rpm for 10 minutes. The upper layer of the solution was collected by using syringe or pipette and was again subjected to the same procedure for 2 to 3 times. Then these collected layers were combined and subjected to dryness under nitrogen streaming. Then the residue remaining in these tubes were mixed with 1 ml of mobile phase and vortexed for half minute and collected in the vials for analysis. After filtration through 0.45 µm non-pyrogenic filter (Minisart, Satorius Stedim Biotech GmbH, Gottingen, Germany), 20 µL sample was injected in Flexar Perkin Elmer HPLC System (Perkin Elmer, USA) equipped with gradient model Flexar pumps system, LC-Shelton CT, 06484 (USA) UV/Visible detector, column oven and degasser (DG-20A5) [16]. A hypersil GOLD $C_{18}$ column (250 x 4.6 mm internal diameter, 5 µm particle size) (Thermo Fisher Scientific Inc.) and a isocratic (acetonitrile:methanol; 35:65) mode was used with the flow rate of 1.5 mL/min. UV spectra were

recorded at 292 nm. The analytes were identified by matching the retention times and spiking the samples with standards whereas quantification was based on an external standard method.

## Total phenolic contents analysis

Total phenolic contents of oils were determined by using UV-Visible spectrophotometer as reported [18]. Briefly, weighed 2.5 g of oil was dissolved in 3 ml of pure hexane followed by 3 ml 80% methanol. Then obtained reaction mixture was subjected to centrifugation at 5000 rpm for about 10 minutes. After this methanol layer was collected in the separate tube and remaining layer was again subjected to same process for 2–3 times and were combined. Now 0.5 ml of this methanol extract and 0.5 ml Folin reagent (Product no F9252; concentration 1.9 N) was taken in test tube. After this 1.5 ml of 20% sodium carbonate and about 7.5 ml of distilled water was also added in each test tube. After this each test tube was covered and kept at room temperature for about 2 hours and absorbance was noted at 755 nm. The percentage of total phenolic contents was determined as mg of Gallic acid per gram of oil extract.

## Total flavonoid contents analysis

Total flavonoid contents of oils were determined by using UV-Visible spectrophotometer as reported [19]. Briefly 0.5 ml of each oil was mixed with methanolic solution of aluminium chloride having concentration of 2% w/v in methanol followed by dilution with 2 to 3 ml of methanol and was kept at room condition for exact 15 minutes and absorbance was noted at wavelength of 430 nm by using spectrophotometer. The percentage of total flavonoids was determined by using quercetin as a standard and concentration was determined by QE/g of oil extract [19].

## Metal ions analysis

The traces of metals in the selected edible oils were determined by following the wet digestion method proposed by Saleh et al. [20] with some modifications. Briefly, 0.5 g of each oil sample was taken in crucible and heated at 350–450 ˚C temperature until whole organic content burned. The obtained ash was dissolved by using digestion mixture of $HNO_3$ and $H_2O_2$ in 6:2 by using hot plate. The temperature was retained at 120 ˚C for approximately 2 h. After cooling, 10 ml of distilled water was added to each sample and mixed. After filtration volume of each sample was made up to 50 ml by using distilled water. Metal contents of final solution were determined by ICP-OES (Teledyne Leeman Labs Prodigy 7, USA).

## Storage of samples

Collected samples of oils (soybean, canola, cottonseed, palm and rapeseed oil) were divided into portions (250 ml in each glass bottle for storage in light and amber glass bottles for storage in dark) and were stored under following storage conditions; stored at room temperature (25–29 ˚C) in the presence and absence of light (daylight), stored at 40 ˚C and 50 ˚C in a glass door hot air incubator in the presence and absence of light (daylight) in the presence and absence of light for a period of three months. Samples were collected after every ten days and analysed for the oxidation parameters.

## Analysis of stored samples

For assessing oxidative stability of these vegetable oils free fatty acid (FFA) value, peroxide value (POV), para-anisidine value (PAV), conjugated dienes (CD) and conjugated trienes (CT) values were determined for each sample.

### Free fatty acid and peroxide value

FFA and PV were determined by following the AOCS official methods Cd 3a-63 and Cd 8–53 respectively [21].

### Para-anisidine value

The PAV was determined by following AOCS official method Cd 18–90. The oil samples diluted in iso-octane were allowed to react with reagent which was prepared by dissolving anisidine in acetic acid (0.25% w/v). After incubation period of 10 minutes colored complexes are produced which show absorbance at 350 nm [21].

### Conjugated dienes and trienes values

The CD as well as CT values was determined by following IUPAC standard protocol in term of extinction coefficient at 232 and 268 nm respectively by using UV-visible spectrophotometer (Lambda 25, Perkin Elmer). For this purpose, 0.1 g of oil sample was weighted and then was diluted with appropriate amount of iso-octane to adjust absorbance within 0.2–0.8. Then after 10 minutes incubation at room temperature absorbance was noted at the specified wavelengths [16].

### Statistical analysis

Three samples of each oil were taken and analysed individually in triplicates. All the results are determined as mean ± standard deviation (SD). Analysis of Variance (ANOVA) was performed to analyse the data for comparison and probability (p) value $\leq 0.05$ was considered significantly different.

## Results and discussion

### Analysis of collected samples

**Fatty acid composition analysis.** Fatty acid composition of selected seed oils is presented in Table 1. All the selected oils were rich in unsaturated fatty acids (UFAs) except the palm oil. The highest percentage of polyunsaturated fatty acids (PUFAs) was found in soybean oil which was 57.75 g/100g and the concentrations of linoleic acid and linolenic acid were found maximum among all the oils i.e. 53.45 g/100g and 4.30 g/100g, respectively. On the other hand, canola possesses 37.42 g/100 g; cottonseed 32.98 g/100 g; palm oil 29.23 g/100 g and rapeseed oil possess 31.01 g/100g PUFAs in its composition. In addition to PUFAs there was considerable percentage of monounsaturated fatty acids (MUFAs) found in these oils. Rapeseed oil possess highest percentage of MUFAs which was 58.87 g/100 g followed by canola, cottonseed, soybean and palm oil which possess 49.32 g/100 g, 45.70 g/100 g, 34.51 g/100 g and 27.01 g/100 g MUFAs in their composition respectively. In addition to PUFAs and MUFAs, saturated fatty acids (SFAs) were also found in all these oils but in lower percentage except palm oil which has 43.25 g/100 g SFAs in its composition. These results of fatty acid composition conclude that soybean oil has highest percentage of PUFAs followed by canola in its composition so these can be easily prone towards oxidative deterioration and will possess lesser oxidative stability as compared to other oils. On the other hand, cottonseed, palm and rapeseed possess lesser percentage of PUFAs in their composition so these oils possess higher oxidative stability as compared to soybean and canola oil.

### Tocopherol and tocotrienol contents analysis

Tocopherols and tocotrienols (collectively known as tocols) profile of selected oils are presented in Table 2 which were separated simultaneously using isocratic elution approach as

**Table 1. Fatty acid composition of selected commercial oils.**

| Fatty acids | Composition (g/100g) | | | | |
|---|---|---|---|---|---|
| | **Soybean** | **Canola** | **Cottonseed** | **Palm** | **Rapeseed** |
| C12:0 | 0.60±0.03[c] | 0.26±0.02[a] | 0.29±0.01[a] | 0.35±0.02[b] | 2.20±0.05[d] |
| C14:0 | Nd | Nd | Nd | 0.65±0.03[c] | Nd |
| C16:1 | 11.14±0.54[b] | 11.40±0.39[b] | 13.20±0.52[c] | 42.25±2.39[e] | 7.01±0.08[a] |
| C18:1 | 34.51±0.99[b] | 49.32±2.53[c] | 45.70±2.47[c] | 27.01±1.24[a] | 25.05±1.21[a] |
| C18:2 | 53.45±2.57[d] | 33.21±0.99[c] | 30.55±1.30[b] | 26.22±1.15[a] | 31.01±1.25[b] |
| C18:3 | 4.30±0.13[c] | 4.21±0.24[c] | 2.43±0.11[a] | 3.01±0.12[b] | Nd |
| C22:1 | Nd | Nd | Nd | Nd | 33.82±0.99[c] |
| ΣSFAs | 11.74 | 11.66 | 13.49 | 43.25 | 9.21 |
| ΣMUFAs | 34.51 | 49.32 | 45.70 | 27.01 | 58.87 |
| ΣPUFAs | 57.75 | 37.42 | 32.98 | 29.23 | 31.01 |

Data is given as mean ±SD; Different letters in superscript represent significant difference among selected oils. ΣSFAs, ΣMUFAs and ΣPUFAs represent percentages of total saturated fatty acids, total monounsaturated fatty acids and total polyunsaturated fatty acids respectively.

shown in Fig 1. All these components were found in different concentrations in these oils. The highest concentration of alpha tocopherol was found in cottonseed oil (459.85 mg/kg) followed by palm oil (345.21 mg/kg). On the other hand, canola (42.36 mg/kg) and soybean (59.05 mg/kg) contain least concentration of alpha tocopherol and was absent in rapeseed oil. The concentration of gamma tocopherol was found in all oils (except cottonseed) in sufficient percentage but highest was found in rapeseed oil which was 1229.45 mg/kg followed by palm (1116.01 mg/kg), canola (658.70 mg/kg) and soybean (244.18 mg/kg). The sigma tocopherol was absent in all oils except canola (35.91 mg/kg). The alpha tocotrienol was absent in all oils but was present in sufficient concentration in soybean oil (253.76 mg/kg). The gamma tocotrienol was present in all oil samples except rapeseed and highest concentration was found in cottonseed oil (678.77 mg/kg) followed by palm (628.15 mg/kg), canola (322.41 mg/kg) and soybean oil (17.61 mg/kg) respectively. Sigma tocotrienol was present only in soybean and rapeseed oil but higher concentration was found in rapeseed oil (954.89 mg/kg). The highest concentration of total tocols was present in rapeseed oil which was 2184.34 mg/kg followed by palm (2089.37 mg/kg). In case of soybean oil concentration of tocols was 1112.11 mg/kg which was higher to some extent than already reported value (913.51 mg/kg) [22,23]. Canola oil revealed total concentration of tocols of 1059.37 mg/kg which was also higher than already reported value

**Table 2. Tocols contents of selected commercial oils.**

| Tocols | Tocols profile of selected oils | | | | |
|---|---|---|---|---|---|
| | Composition mg/kg of oil | | | | |
| | **Soybean** | **Canola** | **Cottonseed** | **Palm** | **Rapeseed** |
| α-tocopherol | 59.05±1.51[b] | 42.36±2.09[a] | 459.85±11.77[d] | 345.21±12.06[c] | Nd |
| γ–tocopherol | 244.18±10.91[a] | 658.70±24.36[b] | Nd | 1116.01±35.94[c] | 1229.45±26.53[d] |
| δ- tocopherol | Nd | 35.91±1.21[a] | Nd | Nd | Nd |
| α- tocotrienol | 253.76±12.21[a] | Nd | Nd | Nd | Nd |
| γ–tocotrienol | 17.61±0.63[a] | 322.41±13.09[b] | 678.77±24.79[c] | 628.15±33.83[c] | Nd |
| δ- tocotrienol | 537.51±20.98[a] | Nd | Nd | Nd | 954.89±40.55[b] |
| Total tocols | 1112.11 | 1059.37 | 1138.62 | 2089.37 | 2184.34 |

Data is given as mean ± SD; Different letters in superscript represent significant (p ≤ 0.05) difference among selected oils.

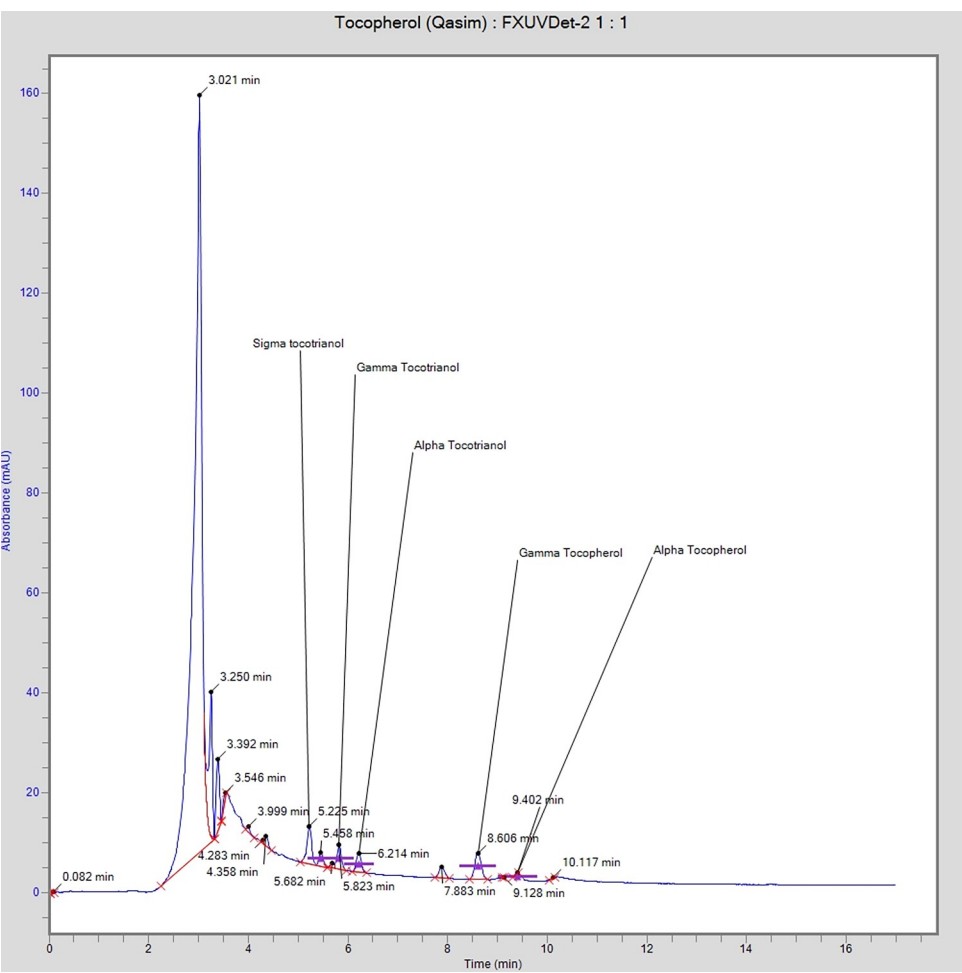

**Fig 1. Typical HPLC chromatogram showing the separation of tocols.**

(822.65 mg/kg) [22,24–26]. Similarly, in case of cottonseed oil this concentration was 1138.62 mg/kg which was higher to some extent than already reported mean value (1010.70 mg/kg) [27]. The concentration of tocols in palm oil (2089.37 mg/kg) was lower than the reported value (3285 mg/kg) in another study because in that case oil used was in crude form and process of purification results in loss of tocopherols [28]. On the other hand, concentration of tocols in rapeseed oil was very higher as compared to literature value (739.80 mg/kg) which is possibly due to different variety as well as processing methods [29]. These concentrations of total tocols also elaborate oxidative stability of these oils. As the highest percentage of total tocols were found in rapeseed oil which elaborates that this oil will possess higher oxidative stability as compared to all other oils. The percentages of total tocols were less in the soybean as well as canola which is also evidence for the least stability of these oils. The values reported were varied to some extent to the value reported in literature which may be due to different processing and purification process of these oils [22,26,30,31].

## Total phenolic and flavonoid contents analysis

The total phenolic contents (TPC) and total flavonoids contents (TFC) of the selected vegetable oils are presented in Table 3. Gallic acid standard curve (y = 0.1012x + 0.1205; $R^2$ = 0.9979) was used to determine the TPC of oils while quercetin standard curve (y = 0.4135x -0.0181; $R^2$

**Table 3. Phenolics and flavonoid contents of selected oils.**

| Total phenolic and flavonoid contents | | | | | |
|---|---|---|---|---|---|
| Analysis | Soybean oil | Canola oil | Cottonseed oil | Palm oil | Rapeseed oil |
| Total phenolic contents (mg GAE/g OE) | 2.93±0.13[a] | 2.84±0.10[a] | 6.67±0.15[b] | 11.10±0.42[c] | 14.44±0.40[d] |
| Total flavonoid contents (mg QE/g OE) | 0.47±0.03[a] | 0.44±0.03[a] | 0.45±0.02[a] | 0.56±0.03[b] | 1.56±0.35[c] |

Data is given as mean ± SD; Different letters in superscript represent significant ($p \leq 0.05$) difference among selected oils. GAE: Gallic acid equivalent; QE: Quercetin equivalent; OE: Oil extract.

= 0.9986) was used for the determination of TFC. TPC and TFC varied from oil to oil. Highest TPC and TFC contents were found in rapeseed oil that were 14.44 mg/g of oil extract, measure as Gallic acid equivalent and 1.56 mg/g of oil extract, measure as quercetin equivalent, respectively. These phenolics were higher to some extent than already reported value (12 mg/g of oil) of high erucic acid variety of rapeseed [32]. Palm oil also revealed higher values for these but less than rapeseed oil. TPC and TFC contents for palm oil were 11.10 mg/g and 0.56 mg/g oil extract respectively. These values were higher than already reported values (i.e., 1.81 mg/g and 0.36 mg/g oil respectively) of different varieties of palm oil [33]. Cottonseed oil also revealed higher values of TPC and TFC which were 6.67 mg/g and 0.45 mg/g oil extract respectively. These values of TPC and TFC were slightly lower and higher than already reported values (8.22 mg/g and 0.05 mg/g) respectively. On the other hand, soybean and canola revealed lower concentration of these phenolic compounds. In case of soybean oil TPC and TFC values were 2.93 mg/g and 0.47 mg/g oil extract. TPC value was slightly lower than already reported value (3.23 mg/g) but TFC value was quite higher than already reported value (0.03 mg/g). Similarly, canola revealed TPC value of 2.84 mg/g which was slightly lower than already reported value (3.01 mg/g) but TFC value was 044 mg/g which was slightly higher than already reported value (0.07 mg/g). These different values are also indication of different oxidative stability of these oils. As canola and soybean oil showed lower values as compared to rapeseed and palm oil this also indicate lower oxidative stability of these oils as TPC and TFC are natural antioxidant and slowdown the process of oxidation [34,35]. These results also concluded that rapeseed oil will possess higher oxidative stability as compared to other oils.

## Metal ion analysis

The data elaborated in Table 4 represents the concentrations of different metals in the selected oils determined by ICP-OES. The highest concentration of metal ions which can contribute towards oxidative deteriorations of vegetable oils were Mg, Fe, Mn and Ca which were ranging from 57.14–114.85 µg/ml, 126.87–460.06 µg/ml, 106.85–538.39 µg/ml and 32.94–78.98 µg/ml, respectively. The highest percentages of these elements were present in canola followed by cottonseed and rapeseed oil. The percentages of these ions were also sufficient in soybean oil. On the other hand, percentages of these metal ions were lower in the palm oil which predicts palm oil will be stable to some extent as compared to other oils. Presences of these metals promote oxidation of oils which is due to lower activation energy required for initiation of autoxidation to about 64 ∼ 103 kJ/mol [7]. These values were close to the already reported values of these metal ions but in some cases these values were different which may be due to soil conditions of these plants, extraction methods as well as treatment of these oils during purification [36].

## Analysis of stored samples

**FFA value.** The FFA contents of various oil stored at different conditions are presented in Fig 2A–2E. The highest percentages of free fatty acids were produced in the samples which

**Table 4. Metal contents of selected oils measured by ICP-OES.**

| Metals | Concentration (µg/mL) | | | | |
|---|---|---|---|---|---|
| | Soybean oil | Canola oil | Cottonseed oil | Palm oil | Rapeseed oil |
| Zn | 0.08±0.01[a] | 0.14±0.01[b] | 0.32±0.03[c] | 0.07±0.01[a] | 0.54±0.04[d] |
| Pb | 0.81±0.07[a] | 1.81±0.07[b] | 6.76±0.23[c] | 0.82±0.05[a] | 7.49±0.26[d] |
| Co | 1.86±0.07[a] | 2.56±0.12[b] | 6.78±0.24[c] | 1.76±0.06[a] | 7.21±0.15[d] |
| Cu | 4.95±0.10[b] | 5.11±0.14[b] | 5.42±0.12[c] | 4.92±0.14[b] | 3.23±0.13[a] |
| Cd | 6.13±0.26[a] | 14.76±0.39[b] | 6.78±0.45[a] | 6.14±0.11[a] | 27.56±0.75[c] |
| Ni | 18.61±0.44[c] | 15.33±0.50[b] | 70.54±2.59[e] | 7.06±0.57[a] | 46.56±2.03[d] |
| Cr | 23.88±1.20[b] | 29.43±1.23[c] | 25.35±1.19[b] | 15.67±0.50[a] | 36.81±1.91[d] |
| Ca | 49.69±2.09[c] | 68.30±2.59[d] | 78.98±2.73[e] | 40.98±1.86[b] | 32.94±1.27[a] |
| Mg | 83.89±3.91[c] | 103.46±4.01[d] | 114.85±4.03[e] | 77.06±2.67[b] | 57.14±2.53[a] |
| Fe | 202.89±8.96[b] | 460.06±10.76[e] | 310.78±7.53[c] | 126.87±3.02[a] | 334.17±9.89[d] |
| Mn | 209.43±9.16[b] | 538.39±24.45[d] | 420.69±18.87[c] | 106.85±3.98[a] | 230.95±12.02[b] |

Data is given as mean ± SD; Different letters in superscript represent significant (p ≤ 0.05) difference among selected oils.

were stored at the highest temperature in the presence of light. Samples stored at 50˚C in the presence and absence of light showed high FFA contents in all the oil samples than samples stored at 40˚C as well as room temperature. Among all oils, the highest percentage of FFA was observed in the soybean oil samples stored at 50˚C temperature in the presence of. After storage time of 90 days FFA value of soybean oil stored at 50˚C and in the presence of light was about 1.65% (as oleic acid) followed by canola (1.49%), cottonseed oil (1.41%), rapeseed oil (1.36%) and palm oil (0.95%). According to recommendation proposed by Codex Alimentarius Commission (1999) the acceptable percentage of FFA for refined oils should be 0.3% after which it should be considered as oxidized [37]. The value of FFA for soybean was higher than the values reported in the literature after 90 days' storage which is due to elevated temperature conditions (0.84 at ambient condition and 1.46 at sunlight storage conditions) [38]. Similar types of results have been observed in other studies which concluded that presence of light and higher temperature results in higher FFA values [39,40]. The higher FFA value of soybean oil as compared to other oils under specified conditions was due to higher percentage of unsaturation in its composition [41].

**Peroxide value.** Peroxide values (PV) are primary indicator of oxidation of oils which also vary according to the storage conditions. The changes in PV according to specified storage condition have been elaborated in Fig 3A–3E. The highest PV was observed for the oil sample of soybean stored at 50˚C in the presence of light followed by the canola and cottonseed oil. After storage period of 90 days' soybean oil sample depict the highest PV of about 40.15 meq/kg followed by canola (35.08 meq/kg), cottonseed (32.39 meq/kg), palm (28.48 meq/kg) and rapeseed oil (24.92 meq/kg). On the other hand, the samples which were stored at 50˚C in the absence of light revealed less increment in POV than 50˚C in the presence of light but higher than other storage conditions. These results were in accordance with the results of previous studies [42]. Also, the POV of soybean oil was higher than the reported value of soybean oil in literature at ambient and sunlight conditions after 90 days' storage (At ambient condition PV was 12.80 meq/kg and sunlight exposure this value was 23.50 meq/kg after 90 days) [38]. Rate of peroxide formation occur at faster rate in the presence of light as well as high temperature because presence of light cause photo-oxidation while higher temperature cause lowering of activation energy as well and promote oxidative deterioration of edible oils at faster rate [40,43]. In addition to this, fatty acid profile of edible oil also influences the oxidation of the

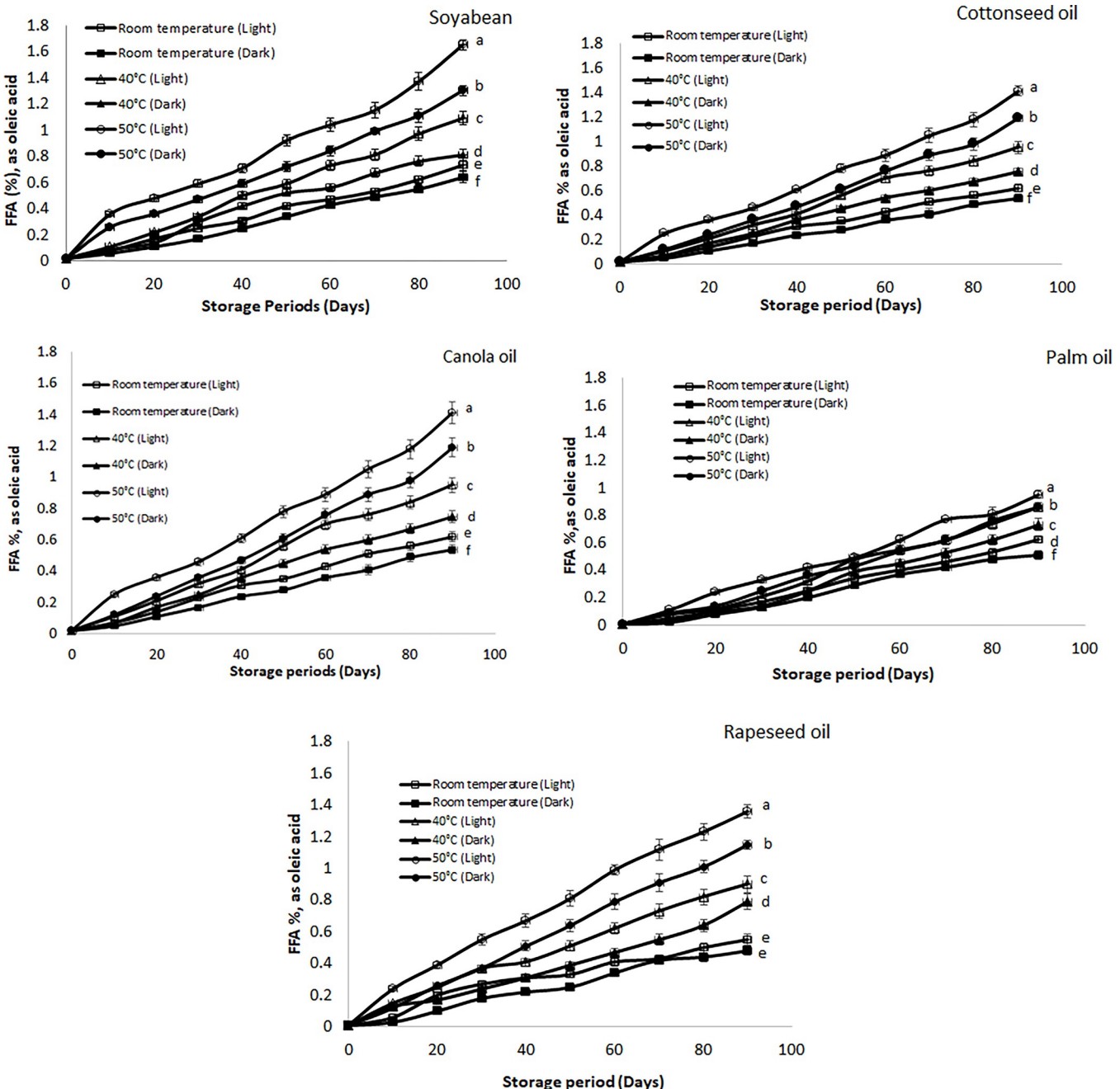

**Fig 2. (a-e):** Effect of different storage conditions on the free fatty acid (FFA) value of selected commercial oils. Different alphabets represent significant ($p \leq 0.05$) different among various storage condition at day 90.

edible oil. As soybean and canola oil are enriched with unsaturated fatty acids so they are at the risk of oxidative deterioration under harsh conditions. Similarly, cottonseed oil also possesses higher percentage of unsaturation in its composition to some extent which prone this oil to same problem of deterioration. On the other hand, rapeseed and palm oil possess less percentage of unsaturated fatty acid as compared to soybean and canola oil due to which they

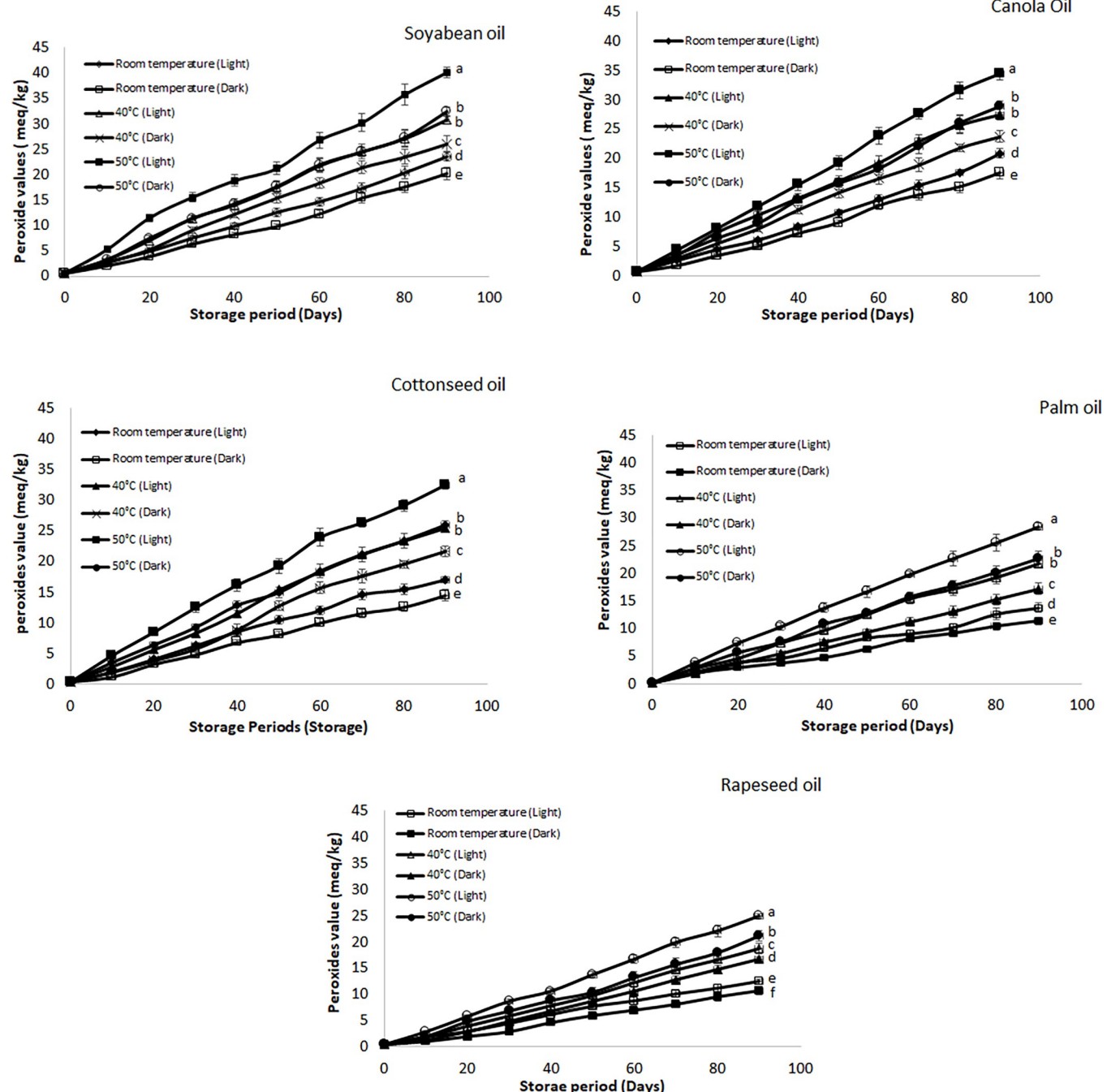

**Fig 3. (a-e):** Effect of different storage conditions on the peroxide value (PV) of selected commercial oils. Different alphabets represent significant (p ≤ 0.05) different among various storage condition at day 90.

possess lower peroxide value [41]. According to the recommendation proposed by Codex Alimentarius Commission (1999) the acceptable value for PV should be equal or less than 10 meq/kg but the samples stored at elevated temperatures showed much higher value which elaborate higher degree of oxidation [37].

**Para-anisidine value.** Para-anisidine value (PAV) mostly represents the percentage of aldehydic products generated in the oils at later stages of oxidation. Like PV, PAV also varies

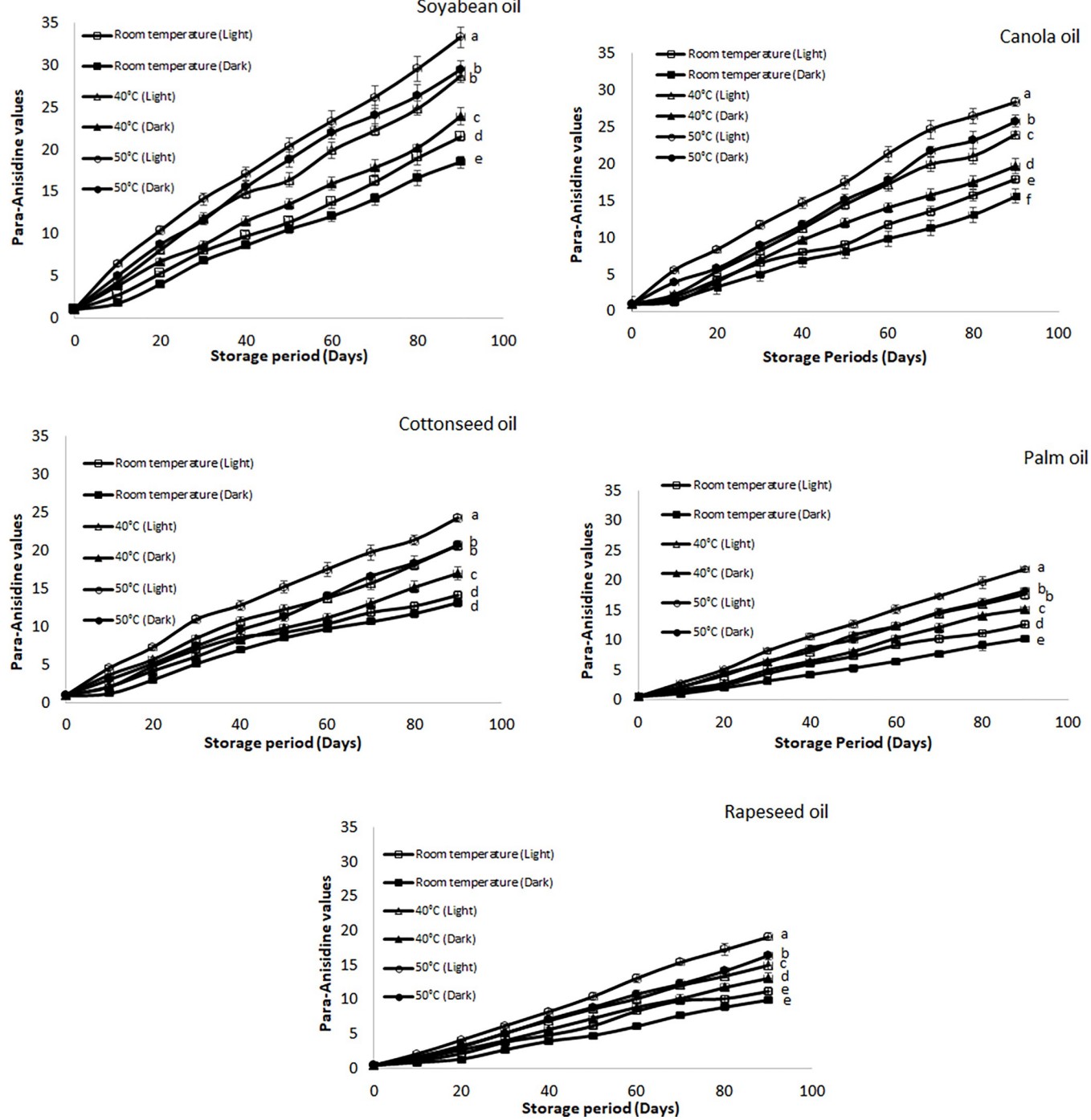

**Fig 4.  (a-e):** Effect of different storage conditions on the para-anisidine value (PAV) of selected commercial oils. Different alphabets represent significant (p ≤ 0.05) different among various storage condition at day 90.

according to the different storage conditions which have been elaborated in the Fig 4A–4E. The PAV value enhanced under each specified storage condition, but highest increment was found for the samples which were stored at higher temperature and presence of light. The results proposed that soybean oil stored at 50°C in the presence of light showed higher PAV

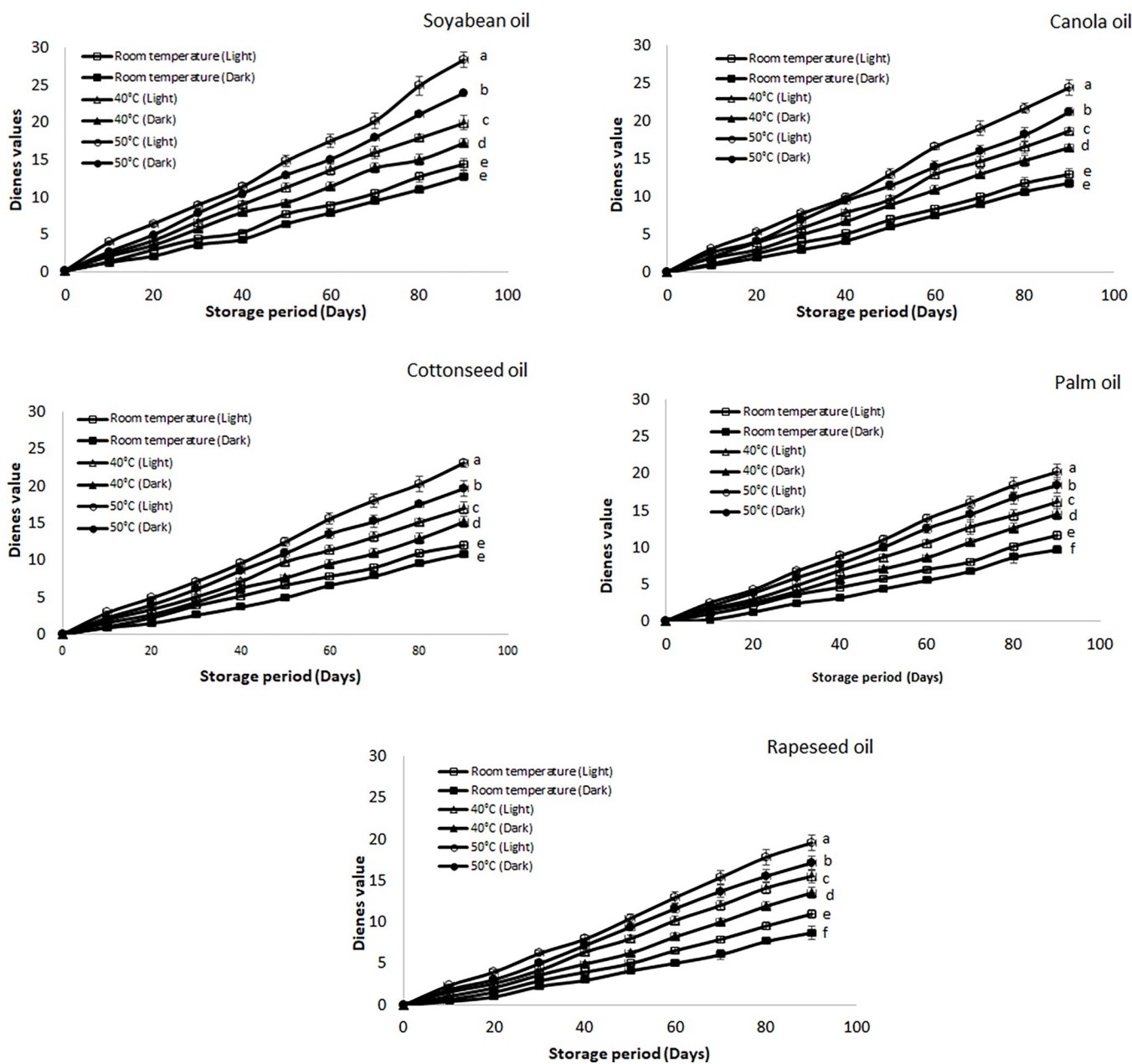

**Fig 5. (a-e):** Effect of different storage conditions on the conjugated dienes (CD) values of selected commercial oils. Different alphabets represent significant (p ≤ 0.05) different among various storage condition at day 90.

value of 33.30 followed by canola (28.41), cottonseed oil (24.21), palm (21.75) and rapeseed oil (19.15). These values were in accordance to already reported results in other studies [44]. Oxidation of soybean oil was observed to be occurred at faster rate as compared to all other oils under different conditions. So, the presence of light and higher temperature causes promotion of deterioration of edible oils or oxidative deterioration. As reported in the literature presence of air, light and frying at higher temperature cause more deterioration [45]. On the other hand, oils which possess higher percentage of unsaturated fatty acids are also prone towards more oxidations as compared to saturated fatty acid containing oils. As soybean oil possess

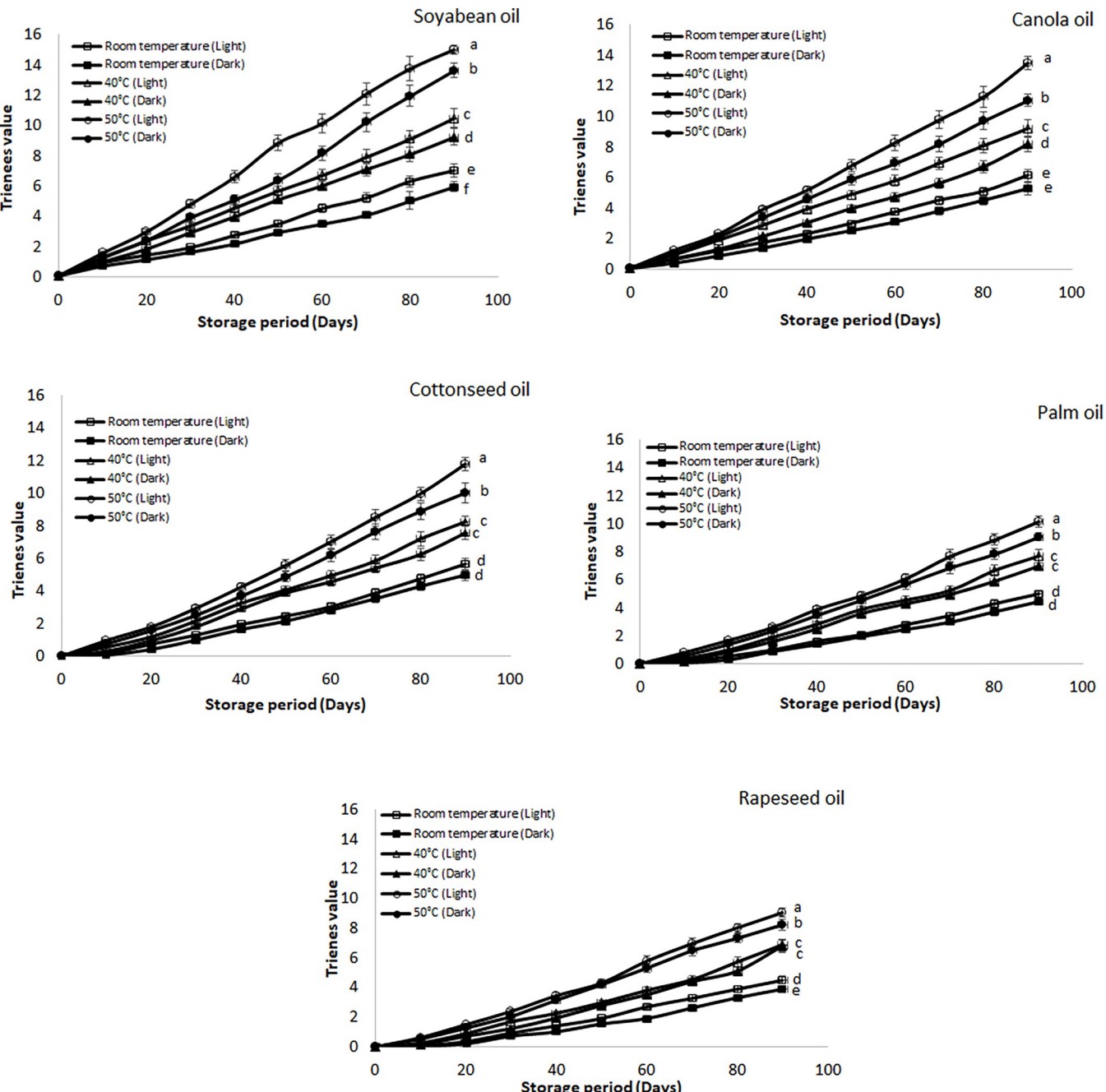

**Fig 6. (a-e):** Effect of different storage conditions on the conjugated trienes (CT) values of selected commercial oils. Different alphabets represent significant (p ≤ 0.05) different among various storage condition at day 90.

higher percentage of unsaturation as compared to other oils it can be deteriorated quite easily and converted into primary products and if conditions become further favourable then they are converted into secondary products [40]. Refined oil should have PAV within the range of 1–10 and above this will indicate oxidation of oil [37].

**Conjugated dienes and trienes values.** The conjugated products are mostly generated in the oil samples because of bond displacement reactions and are mostly represented as

conjugated dienes and trienes (CD and CT). The results of CD and CT of these selected oils have been elaborated in Figs 5A–5E and 6A–6E, respectively which were stored under different storage conditions. Like other types of oxidation products, the percentage of both CD and CT were higher in the samples which were stored at higher temperature as well as in the presence of light. The highest value of CD was found for soybean oil (28.41) followed by canola (24.42), cottonseed (23.10), palm (20.23) and rapeseed oil (19.52) stored at 50˚C temperature and presence of light respectively. Like CD the highest values of CT were found for soybean oil stored at 50˚C and presence of light which was 15.02 followed by canola (13.51), cottonseed (11.78), palm (10.18) and rapeseed oil (9.01) respectively. The values of these products found for soybean at 50˚ C in the presence of light were higher than reported values of soybean which was due to presence of light and elevated temperature [38]. On the other hand, most of these oils are enriched in unsaturation so there are more chances for the generation of oxidation products. As the unsaturation in the oil increases more are the chances of the conjugated products formation [46]. Among all the selected oil there was highest percentage of unsaturation in the soybean oil followed by canola and cottonseed oil. So, the highest percentages of these products were found in the soybean oil. On the other hand, palm and rapeseed showed lowest percentage of these products because these oils have lower percentage of unsaturation in their composition as compared to soybean, canola and cottonseed oil [44].

## Conclusion

The present study was conducted to determine fatty acid profile, tocols composition, phenolic contents and oxidative stability of fully refined, bleached and deodorized soybean, canola, cottonseed, palm and rapeseed oil under different storage condition which include room temperature in the presence and absence of light, 40˚C in the presence and absence of light and 50˚C in the presence and absence of light. Soyabean oils has the highest contents of PUFA and thus more prone to oxidation. Canola oil has the highest contents of oleic acid thus showed more oxidative stability. Oil quality analysis showed proved that the higher percentage of unsaturation, lower percentage of antioxidants, traces of metals, the higher temperature and presence of light prone oils towards more oxidative deterioration. All the samples stored under different storage conditions revealed different extent of oxidation but higher extent was found for the samples which were stored at 50˚C in the presence of light because values of all the oxidation parameters were higher for these samples as compare to others. As a result of this multi-parameter analysis the better plan could be developed for preservation of oil without deterioration of quality of edible oils.

## Supporting information

**S1 File. Standard curves for total phenolic contents (TPC) and total flavonoids Contents (TFC).**
(XLSX)

**S2 File. Graphs for the oxidative stabilities of seed oils.**
(XLSX)

## Acknowledgments

The author acknowledges the services provided by the Central Hi-Tech Lab, Government College University Faisalabad, Pakistan, for the instrumental analysis.

## Author Contributions

**Conceptualization:** Iqbal Ahmed, Abdullah Ijaz Hussain.

**Investigation:** Muhammad Furqan Farooq.

**Methodology:** Iqbal Ahmed, Hira Zulfiqar.

**Project administration:** Shahzad Ali Shahid Chatha.

**Software:** Neelam Iftikhar, Muhammed Ali Alshehri.

**Supervision:** Abdullah Ijaz Hussain.

**Validation:** Neelam Iftikhar, Syed Makhdoom Hussain.

**Writing – original draft:** Iqbal Ahmed, Muhammad Furqan Farooq, Hira Zulfiqar.

**Writing – review & editing:** Shafaqat Ali, Muhammed Ali Alshehri, Khalid A. Al-Ghanim.

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
