## [Decision Letter · Decision Letter 0]

17 Jun 2024

PONE-D-24-22445Effect of storage plans on the oxidative stability and nutritional quality of selected commercially available seed oilsPLOS ONE

Dear Dr. Ijaz,

Thank you for submitting your manuscript to PLOS ONE. After careful consideration, we feel that it has merit but does not fully meet PLOS ONE’s publication criteria as it currently stands. Therefore, we invite you to submit a revised version of the manuscript that addresses the points raised during the review process.

We look forward to receiving your revised manuscript.

Kind regards,

Mojtaba Kordrostami, Ph.D.

Academic Editor

PLOS ONE

Journal Requirements:

2. In the online submission form you indicate that your data is not available for proprietary reasons and have provided a contact point for accessing this data. Please note that your current contact point is a co-author on this manuscript. According to our Data Policy, the contact point must not be an author on the manuscript and must be an institutional contact, ideally not an individual. Please revise your data statement to a non-author institutional point of contact, such as a data access or ethics committee, and send this to us via return email. Please also include contact information for the third party organization, and please include the full citation of where the data can be found.

Reviewers' comments:

Reviewer's Responses to Questions

**Comments to the Author**

1. Is the manuscript technically sound, and do the data support the conclusions?

Reviewer #1: Partly

Reviewer #2: Yes

2. Has the statistical analysis been performed appropriately and rigorously? 

Reviewer #1: Yes

Reviewer #2: Yes

3. Have the authors made all data underlying the findings in their manuscript fully available?

Reviewer #1: Yes

Reviewer #2: Yes

4. Is the manuscript presented in an intelligible fashion and written in standard English?

Reviewer #1: Yes

Reviewer #2: Yes

5. Review Comments to the Author

Reviewer #1: Dear Authors

This research article aimed to investigate the effect of storage conditions on the oxidative stability of vegetable oils. Generally, the manuscript (MS) is fairly written and organized. MS contains some useful information. The methodology and data collection correspond to the objectives. However, to make this MS meet the standard of the PLOS ONE Journal, some more information and revisions of the MS are required.

1. The Title may be revised as the nutritional compositions of the oils were only determined in raw materials before storage. This would suggest “Nutritional quality of selected seed oils and effect of storage conditions on their oxidative stability”

2. The abstract should be revised. Some important information must be covered and reflect the main findings according to the objectives.

-Lines 21, the seed oils used in this study should be specified, what are they?

-Short information about methodology must be included as the abstract should be stand-alone.

-Line 23, should be mentioned the values of all seed oils.

- Reorganize the writing, the results in the abstract should be written in the same order as in the results section.

Lines 34-35 should be revised, please see the details and comments provided in the conclusion section.

3. Introduction section:

3.1 Lines 76-82 should be deleted. It has already been mentioned in the methodology.

3.2 The literature review is insufficient. A background information and literature review on the effect of improper environmental conditions on oil quality and chemical compositions reported by previous studies are required.

4. Materials and methods

4.1 Lines 145-151, please revise and detail the storage conditions. The room temperature should be in range, not a single temperature. How to get the oils to explode to light and absent to light. What kind and intensity of light? How to control temperature during storage at high temp.

5. Results and discussion

-More discussion on section FA composition, tocopherols, TPC, and TFC are required. The values of these parameters obtained in this study could be both higher and lower than previous studies or even similar to other studies, so authors need to discuss all sides. More citations are required.

- From the values of FFA, PV, and PAV, authors should also discuss the standard values of these parameters, as long as these values obtained do not exceed the standard values, these would be accepted.

-In Figures 1, 2, and 3, the error bars or superscripts should be inserted to indicate statistical differences.

6. The conclusion should be revised. In lines 339-343, the authors did not determine or compare the oxidative stability values among the oil types, so authors should not conclude whether SBO is the most unstable, instead, authors should focus on answering the main objectives. If authors would like to conclude or provide this information, authors should statistically compare these values among the oil types.

Reviewer #2: The authors did lots of works, and the results were meaningful. some suggestions were presented:

1) Total Phenolic Contents: one time was not enough to fully extract the phenolic compounds by 80% methanol and hexane. three times might be better. in addation, the concentration of Folin was missing. detailed information maybe important for other researchers to repeat your tests and data.

2)Tocopherol and Tocotrienol Contents: the determination conditions by HPLC should be presented clearly. If possible, please provided the HPLC chromatogram.

6. PLOS authors have the option to publish the peer review history of their article (what does this mean?). If published, this will include your full peer review and any attached files.

Reviewer #1: No

Reviewer #2: **Yes: **Liyou Zheng

---

## [Author Response · Author response to Decision Letter 0]

6 Jul 2024

The manuscript has been carefully revised and improved in all aspects including its grammar, clarity and language. All the comments/reservations have been fully addressed. The supplementary material is added and uploaded. The uploaded manuscript is now without any highlights and a separate manuscript is uploaded with track changes as well. . 

Reviewer #1

Comment 1: The Title may be revised as the nutritional compositions of the oils were only determined in raw materials before storage.

Author’s response: We appreciate the reviewer's insightful suggestion and revised the title which is now more reflection of the results. The revised title is “Nutritional quality of selected commercially available seed oils and effect of storage conditions on their oxidative stability”

Comment 2: The abstract should be revised. Some important information must be covered and reflect the main findings according to the objectives. Lines 21, the seed oils used in this study should be specified, what are they? Short information about methodology must be included as the abstract should be stand-alone. Line 23 should be mentioned the values of all seed oils. Reorganize the writing, the results in the abstract should be written in the same order as in the results section. Lines 34-35 should be revised please see the details and comments provided in the conclusion section.

Author’s response: We appreciate the reviewer's all insightful suggestions regarding improvement of abstract. The abstract is revised as per guidelines and the methodology and results are added in the abstract now as per suggestion.

Comment 3: Introduction section

3.1 Lines 76-82 should be deleted. It has already been mentioned in the methodology. 3.2 The literature review is insufficient. A background information and literature review on the effect of improper environmental conditions on oil quality and chemical compositions reported by previous studies are required.

Author’s response: The introduction section is revised, and the more relevant literature is added. The repeated sentences are also deleted as suggested. All the changes have been marked with blue font.

Comment 4: Materials and methods: 4.1 Lines 145-151, please revise and detail the storage conditions. The room temperature should be in range, not a single temperature. How to get the oils to explode to light and absent to light? What kind and intensity of light? How to control temperature during storage at high temp?

Author’s response: The section is revised as per guidelines. The detail of sample storage is added in the section. The range of room temperature (25-29 oC) is added. Samples were exposed to daylight in a transparent glass container while for the storage in dark, the samples were kept in amber glass bottles and covered with aluminum foil to avoid light. Moreover, the storage temperatures were maintained in glass door hot air ovens.

Comment 5: Results and discussion: More discussion on section FA composition, tocopherols, TPC, and TFC are required. The values of these parameters obtained in this study could be both higher and lower than previous studies or even like other studies, so authors need to discuss all sides. More citations are required. From the values of FFA, PV, and PAV, authors should also discuss the standard values of these parameters, as long as these values obtained do not exceed the standard values, these would be accepted. In Figures 1, 2, and 3, the error bars or superscripts should be inserted to indicate statistical differences.

Author’s response: Thank you so much for your valuable suggestions. We have updated discussion of results of the above-mentioned sections mainly fatty acid composition, tocopherols TPC and TFC. Now results have been elaborated separately and compared with previous studies. Also, more citations have been added which support our discussion of results. Furthermore, values of oxidation parameters have been discussed in comparison of standard values or acceptable range of values which indicate extent of oxidation or quality of oils. Error bars have been inserted in all the figures and the ANOVA was applied and different alphabets are mentioned in the figures showing significant (p ≤ 0.05) different among various storage conditions at day 90.

Comment 6: Conclusion

 The conclusion should be revised. In lines 339-343, the authors did not determine or compare the oxidative stability values among the oil types, so authors should not conclude whether SBO is the most unstable; instead, authors should focus on answering the main objectives. If authors would like to conclude or provide this information, authors should statistically compare these values among the oil types.

Author’s response: We have revised this section accordingly and conclude only about main objectives instead of comparison among oils.

Reviewer #2

Comment 1: 

Total Phenolic Contents: one time was not enough to fully extract the phenolic compounds by 80% methanol and hexane. Three times might be better. In addition, the concentration of Folin was missing. Detailed information may be important for other researchers to repeat your tests and data.

Author’s response: Thank you so much for noticing this point. The 3 batches of extractions were performed and combined but did not mention in the text. Now this information is added to the text. here so, we have made the correction about this mistake. Moreover, the concentration of Folin reagent used during this study has also been updated in the main text.

Comment 2: 

The determination conditions by HPLC should be presented clearly. If possible, please provided the HPLC chromatogram 

Author’s response: A detailed method and HPLC condition is given now in the text and a typical chromatogram showing separation of tocols has been provided in the main manuscript.

---

## [Decision Letter · Decision Letter 1]

17 Jul 2024

Nutritional quality of selected commercially available seed oils and effect of storage conditions on their oxidative stability

PONE-D-24-22445R1

Dear Dr. Ijaz,

We’re pleased to inform you that your manuscript has been judged scientifically suitable for publication and will be formally accepted for publication once it meets all outstanding technical requirements.

Kind regards,

Mojtaba Kordrostami, Ph.D.

Academic Editor

PLOS ONE

Additional Editor Comments (optional):

Reviewers' comments:

Reviewer's Responses to Questions

**Comments to the Author**

1. If the authors have adequately addressed your comments raised in a previous round of review and you feel that this manuscript is now acceptable for publication, you may indicate that here to bypass the “Comments to the Author” section, enter your conflict of interest statement in the “Confidential to Editor” section, and submit your "Accept" recommendation.

Reviewer #1: All comments have been addressed

Reviewer #2: All comments have been addressed

2. Is the manuscript technically sound, and do the data support the conclusions?

Reviewer #1: Yes

Reviewer #2: Yes

3. Has the statistical analysis been performed appropriately and rigorously? 

Reviewer #1: Yes

Reviewer #2: Yes

4. Have the authors made all data underlying the findings in their manuscript fully available?

Reviewer #1: Yes

Reviewer #2: Yes

5. Is the manuscript presented in an intelligible fashion and written in standard English?

Reviewer #1: Yes

Reviewer #2: Yes

6. Review Comments to the Author

Reviewer #1: The title, abstract, methodology, results, and discussion have now been improved. Some references were updated. According to these changes, the authors appear to have adequately amended the comments made by the reviewer. Thank you for your effort in amending the manuscript.

Reviewer #2: (No Response)

7. PLOS authors have the option to publish the peer review history of their article (what does this mean?). If published, this will include your full peer review and any attached files.

Reviewer #1: No

Reviewer #2: **Yes: **Liyou Zheng

---

## [Editor Report · Acceptance letter]

9 Aug 2024

PONE-D-24-22445R1 

PLOS ONE

Dear Dr. Ijaz Hussain, 

I'm pleased to inform you that your manuscript has been deemed suitable for publication in PLOS ONE. Congratulations! Your manuscript is now being handed over to our production team.

Kind regards, 

on behalf of

Dr. Mojtaba Kordrostami 

Academic Editor

PLOS ONE